# Hydration and Mechanical Properties of High-Volume Fly Ash Cement under Different Curing Temperatures

**DOI:** 10.3390/ma17194716

**Published:** 2024-09-26

**Authors:** Young-Cheol Choi

**Affiliations:** Department of Civil and Environmental Engineering, Gachon University, Seongnam 13120, Gyeonggi-do, Republic of Korea; zerofe@gachon.ac.kr; Tel.: +82-31-750-5721

**Keywords:** curing temperature, high-volume fly ash, hydration, compressive strength

## Abstract

This study aimed to investigate the effects of different curing temperatures on the hydration and mechanical properties of high-volume fly ash (HVFA) concrete. The key variables were curing temperature (13 °C, 23 °C, 43 °C) and fly ash (FA) content (0%, 35%, 55%). The hydration characteristics of HVFA cement were examined by evaluating the setting time and heat of hydration under different curing temperatures. The mechanical properties of HVFA concrete were analyzed by preparing concrete specimens at various curing temperatures and measuring the compressive strength at 7, 28, 56, and 91 days. The results indicated that concrete with high FA content was more sensitive to curing temperature compared to ordinary Portland cement.

## 1. Introduction

With industrial development, global demand for concrete continues to rise. While concrete is cost-effective and durable, it consumes high energy and emits significant greenhouse gases [1,2]. The cement sector, responsible for 75–90% of emissions, contributes 9% to global CO_2_ emissions [3,4]. Efforts to reduce these emissions include using supplementary cementitious materials (SCMs) to reduce cement consumption [5,6]. Large structures such as skyscrapers and bridges also increase the demand for mass concrete, which can lead to cracks due to heat from hydration [7]. Replacing cement with SCMs mitigates this risk, lowers heat generation, and enhances long-term strength and durability, although SCMs provide lower early-age strength [8,9,10,11,12].

High-volume fly ash (HVFA) concrete has emerged to reduce CO_2_ emissions further [7]. Recent research has focused on its physical and hydration properties. Factors like cement composition, water–cement ratio, mineral additives, and temperature influence hydration [13]. Low temperatures slow hydration and delay property development, while high temperatures accelerate it [14,15]. Studies have examined how fly ash (FA) and cement composition affect workability and strength [16,17,18]. Liu et al. found that low temperatures slow hydration and reduce mechanical properties [14]. Han et al. found FA accelerates hydration by providing nucleation sites [19]. Narmluk et al. noted FA initially delays hydration at low temperatures but accelerates it later, with more pronounced effects at higher FA content [20]. Elkhadiri and Puertas showed that high temperatures improve early strength but harm long-term properties, while low temperatures lead to slower early strength gains that increase over time [21]. Parry-Jones found a linear relationship between curing temperature (20–55 °C) and compressive strength, with a drop at 80 °C [22]. Escalante-García and Sharp found GGBS and volcanic ash accelerated hydration, while FA delayed C_2_S hydration at high temperatures, making FA–cement mixtures more sensitive to temperature [19,23,24]. According to the literature reviews on this topic, the effects of FA on cement hydration have been extensively studied. Cementitious materials containing FA are more temperature-sensitive compared to other SCMs, such as GGBS, and their hydration process differs from that of conventional Portland cement paste. However, research on the early hydration characteristics and long-term mechanical properties of FA cement, particularly HVFA cement or concrete, under varying temperatures is still quite limited. Therefore, this study conducted a comprehensive investigation on high-volume fly ash concrete, examining its early hydration reactions and long-term mechanical properties under various curing temperatures.

This study conducted an experimental investigation to evaluate the effect of curing temperature on the hydration and mechanical properties of HVFA concrete. The main variables were curing temperature (13 °C, 23 °C, 43 °C) and FA content (0%, 35%, 55%). Additionally, the effect of FA particle size was studied by producing fine FA using a ball mill. To assess the mechanical properties under different curing temperatures, concrete specimens were prepared, and their compressive strength was measured and analyzed at various test ages. Paste specimens were also created to determine setting times, and an isothermal calorimeter was utilized to analyze the early hydration characteristics at different curing temperatures.

## 2. Materials and Method

### 2.1. Materials

In this study, ordinary Portland cement (OPC) from S Corporation in South Korea was used as the binder. The Blaine fineness and density of the OPC were measured at 355 m^2^/kg and 3.25 g/cm^3^, respectively. The FA used was refined FA from an H power plant in South Korea, with a Blaine fineness of 374 m^2^/kg and a density of 2.42 g/cm^3^. The oxide compounds were determined by XRF and are shown in Table 1. The FA contained CaO: 4.9%, SiO_2_: 51.3%, Al_2_O_3_: 23.7% and Fe_2_O_3_: 7.8%, classifying it as class F according to ASTM C 618. Crushed aggregates were used for both coarse and fine aggregates, with the maximum size of the coarse aggregate being 25 mm. The densities of the coarse and fine aggregates were 2.6 g/cm^3^ and 2.5 g/cm^3^, respectively.

### 2.2. Mixture Proportions

Table 2 outlines the mixture proportions for each specimen, with the main variables being the FA substitution rate (35%, 55%), fineness (374 m^2^/kg, 568 m^2^/kg), and curing temperature (13 °C, 23 °C, 43 °C). To ensure that all specimens reached similar 28-day compressive strength, different amounts of binder, aggregate, and water–binder ratios were used in the mix designs, as detailed in Table 2.

### 2.3. Methods

The morphology of the raw materials was analyzed using a field emission scanning electron microscope (SU 8220, Hitachi High-Technologies, Tokyo, Japan) under conditions of an accelerating voltage of 15 kV and a working distance of 15.2 mm. The samples were dried in an oven at 60 °C, coated with platinum, and then SEM images were taken. The particle size distribution of the binder materials was determined using a Beckman Coulter LS 230 laser diffraction instrument (Beckman Coulter, Brea, CA, USA). To prevent hydration reactions, OPC and FA were dispersed in an isopropyl alcohol solution. The qualitative mineralogical composition of OPC and FA was determined using an X’Pert Pro MPD X-ray diffractometer (Malvern Panalytical, Malvern, UK) with Cu K-alpha radiation. The XRD analysis was conducted at a scanning rate of 5°/min and a step size of 0.02°, covering a 2θ range of 5° to 75°. Prior to analysis, the samples were dried in an oven at 60 °C for 48 h.

The setting time was determined for paste specimens in a chamber under various temperature conditions (13 °C, 23 °C, 43 °C) according to ISO 9597 [25]. An automatic PA8 tester (ACMEL LABO, Saint Pierre du Perray, France) was used to measure the penetration depth of the needle over time, which was then used to calculate the setting time. To evaluate the heat of hydration of the binder at different curing temperatures, an isothermal calorimeter was employed. The study utilized a TAM Air isothermal calorimeter (TA Instruments, New Castle, DE, USA) with eight test channels. According to the mixture proportions outlined in Table 2, excluding aggregates, paste was prepared by mixing the binder and distilled water for each experimental condition. The OPC, FA, and water were stored in the same chamber as the target curing temperature conditions one day prior to use. After mixing, 4 to 5 g of paste was placed into 20 mL glass vials and installed in the isothermal calorimeter. The weight of the glass vials was measured before and after adding the paste to determine the exact amount of paste used. The TAM Air’s temperature was set to the desired curing temperature (13 °C, 23 °C, 43 °C) 24 h before testing and maintained consistently throughout the experiment. The heat flow of each specimen was continuously and automatically measured for 72 h. Experiments for each variable were performed in triplicate, and the average value was taken as the final result.

The compressive strength of the concrete specimens was assessed according to ASTM C39. All materials needed for the mix were stored at 23 °C in a temperature- and humidity-controlled room until batching. Concrete specimens were prepared using the proportions in Table 2 and cast into cylindrical molds measuring 100 mm in diameter and 200 mm in height. The specimens were cured for 24 h in a chamber set at 23 ± 3 °C and a relative humidity above 95%, after which they were demolded. The specimens were then water-cured at specified temperatures (13 °C, 23 °C, 43 °C). Compressive strength tests were performed at 7, 28, 56, and 91 days. To ensure reliability, three specimens were tested for compressive strength at each age, and the average value was used as the final result.

## 3. Results and Discussion

### 3.1. Setting Behavior of Specimens

Figure 1 displays the SEM images of OPC and FA. The OPC particles are angular in shape, whereas the FA particles are mostly spherical, with some irregularly shaped particles also present.

The XRD patterns of OPC and FA are depicted in Figure 2. The primary components of OPC are identified as C_3_S, C_2_S, C_3_A, C_4_AF, and gypsum. Additionally, calcite is detected due to the addition of limestone powder during cement production. The crystalline phases present in FA, as shown in Figure 2, include quartz, mullite, and magnetite. Moreover, the presence of an amorphous phase is indicated by a hump in the 2θ range of 18° to 29°. FA consists of both crystalline and amorphous phases, which vary based on the operating conditions of the power plant and the type of fuel used.

In this study, FA was ground using a ball mill to assess the impact of particle size on the hydration and mechanical properties of cement. The Blaine fineness of the ground FA (FA-F) is 568 m^2^/kg. Figure 3 shows the particle size distributions of OPC, FA, and FA-F. Particle sizes of the raw materials were measured using laser diffraction, assuming spherical particles. The average particle diameters of OPC, FA, and FA-F are 17.56 μm, 12.37 μm, and 9.03 μm, respectively.

Figure 4 displays the initial and final setting times at curing temperatures of 13 °C, 23 °C, and 43 °C. As shown in the figure, both initial and final setting times increase with higher curing temperatures. For the plain and F35 specimens, the setting times significantly increased at 13 °C, while the F55 and F55F specimens showed a more gradual increase. At 43 °C, all specimens exhibited similar setting times, with a slight decrease as the FA content increased. FA generally provides a substantial amount of silicate ions to the pore solution during the early hydration phase of cement and adsorbs calcium ions on its surface. This property of FA lowers the Ca/Si ratio in the pore solution within the matrix. The lower Ca/Si ratio in the C-S-H of FA cement paste can slow down the transition to the more stable C-S-H compared to pure cement paste with a higher Ca/Si ratio [26,27,28]. As a result, the induction period is extended, delaying the setting time of FA-blended cement paste [20].

### 3.2. Effect of Curing Temperature on Heat of Hydration

Figure 5 shows the heat rate results for each variable at different curing temperatures. In all instances, as the curing temperature decreased, the induction period tended to lengthen, reflecting the setting time results shown in Figure 4. This trend was more pronounced with higher FA substitution rates [29,30,31], due to the reduced C_3_S content in OPC with increasing FA content. Furthermore, higher curing temperatures resulted in increased second peaks for all specimens, indicating that low-temperature curing suppresses nucleation and growth kinetics, whereas high-temperature curing enhances them [32]. For the plain specimen, the acceleration period began after 5.80 h, 2.52 h, and 1.78 h at curing temperatures of 13 °C, 23 °C, and 43 °C, respectively. This is linked to the formation of calcium silicate hydrates, which significantly impact the stiffness and strength development of cementitious materials. As the FA substitution rate increased, the induction period extended, delaying the onset of the acceleration period. For the F55 specimen, the acceleration period began after 21.98 h, 9.82 h, and 4.95 h at curing temperatures of 13 °C, 23 °C, and 43 °C, respectively, indicating a longer induction period regardless of the curing temperature. At 13 °C, higher FA substitution rates led to a more noticeable hydration delay. For F55F, which included finely ground FA-F, hydration was accelerated compared to F55, resulting in a shorter induction period and an earlier start of the acceleration period. This is because finely ground FA provides additional nucleation sites, enhancing the filler effect.

Figure 6 presents the second peak results for specimens at different curing temperatures. As illustrated, the second peak occurs earlier with increasing curing temperatures. This trend is especially noticeable in specimens with 55% FA by cement weight, specifically F55 and F55F. At a curing temperature of 13 °C, the second peak for F55 occurs at 36.03 h, which is 17 h later than that of the plain specimen. However, at 43 °C, the second peak for F55 appears at 8.53 h, only 2.66 h later than the plain specimen. This indicates that a high FA content in cement significantly delays the hydration reaction at lower temperatures compared to higher temperatures. For F55F, the second peak occurs earlier than for F55 at curing temperatures of 13 °C and 23 °C, suggesting that the finer FA particles accelerate the hydration reaction [33].

Figure 7 presents the heat hydration results for specimens at various curing temperatures. As expected, the heat hydration for the plain specimen was the highest at each curing temperature compared to other variables, with a tendency for heat hydration to decrease as the FA content increased. All specimens showed inhibited hydration reactions at lower curing temperatures and accelerated reactions at higher curing temperatures.

Figure 8 shows the 72 h cumulative heat hydration for each specimen at different curing temperatures. For the plain specimen, the heat hydration at 23 °C was 270.0 J/g. Relative to this, the heat hydration decreased by 19% at 13 °C and increased by 28% at 43 °C. The 72 h heat hydrations for F35 and F55 at 23 °C were 72.0% and 53.0% compared to that of the plain specimen, respectively, indicating a proportional relationship to the FA substitution rate. For F55F, the cumulative heat release was 57.2%, slightly higher than that of F55, likely due to the seeding effect of the finer FA-F particles. At 13 °C and 43 °C, the heat hydration of F35 was 79.7% and 138.5%, respectively, compared to that at 23 °C, showing a similar trend to the plain specimen at low temperatures but a slightly higher increase at high temperatures. For F55, the heat hydration at 13 °C was 73.0% of that at 23 °C, while at 43 °C, it increased significantly to 144.6%. F55F showed a similar pattern to F55. These results indicate that the curing temperature significantly affects the hydration reaction when a high amount of FA is used [34,35].

### 3.3. Compressive Strength Results

Figure 9 shows the compressive strength results of specimens cured at 23 °C. As depicted, the 28-day compressive strength was nearly identical for all variables, indicating that the mix design was effective as planned in Section 2.2. At 7 days, all FA-containing specimens had lower compressive strength compared to the plain specimen. The 7-day compressive strengths of F35 and F55 were 90.4% and 78.9% of the plain specimen, respectively, due to the reduced C_3_S content in the binder from replacing OPC with FA, leading to fewer early hydration products. The 7-day compressive strength of F55-F was 85.4% of the plain specimen and 8.2% higher than that of F55, suggesting that finely ground FA can help mitigate the early strength reduction typically seen in FA-blended cementitious materials [33].

The compressive strength results at 56 and 91 days differed from those at 7 days. F55 exhibited compressive strength similar to the plain specimen, while F35 and F55F showed higher compressive strengths than the plain specimen. Notably, F55F demonstrated compressive strengths 6.6% and 11.9% higher than the plain specimen at 56 and 91 days, respectively. This difference is due to the pozzolanic reaction of FA. Typically, FA does not significantly react within the first 7 days of hydration [36]. After 7 days, the hydration of C_3_S in cement increases the amount of Ca(OH)_2_ around the cement particles. The FA particles, present with the C_3_S particles, then come into contact with water, causing alkali ions (Na^+^, K^+^) to leach out and the FA surface to become negatively charged. This attracts cations (H^+^ or H_3_O^+^) in the solution, leading to surface reactions. Continuous dissolution of alkali ions forms amorphous Si and Al-rich regions on the FA surface, leading to the release of SiO_4_^4−^ and AlO_2_^−^ ions. The negatively charged FA surface reacts with Ca^2+^ ions from the Ca(OH)_2_ produced during C_3_S hydration, forming a C-S-H layer around the FA. This newly formed C-S-H is predominantly a reticular type II C-S-H. Additionally, this process generates other cementitious compounds, such as calcium aluminate hydrates (C_4_AH_13_, C_2_AH_8_) and calcium aluminosilicate hydrates (C_2_ASH, C_3_AS_3_-C_3_AH). This pozzolanic reaction reduces the porosity of the cement matrix, enhancing its durability [37,38,39].

Figure 10 presents the compressive strength results of specimens cured at 13 °C over time. The trend in compressive strength development for specimens cured at this lower temperature differed from those cured at room temperature. All specimens exhibited a decrease in compressive strength compared to those cured at 23 °C, with the reduction varying according to the FA content. The average compressive strength ratios at 7 and 28 days for specimens cured at 13 °C versus 23 °C were 0.89 for plain, 0.77 for F35, and 0.68 for F55, indicating a greater reduction in compressive strength with higher FA content under low-temperature curing. F55F exhibited a similar ratio of 0.70, comparable to that of F55. As shown in Figure 8, the compressive strength of plain increased gradually after 28 days. However, specimens with FA showed a more significant increase in compressive strength after 28 days compared to plain. The average compressive strength ratios at 56 and 91 days for specimens cured at 13 °C versus 23 °C were 0.90 for plain, 0.90 for F35, and 0.84 for F55. The reduction in long-term compressive strength for FA-containing specimens decreased, likely due to the pozzolanic reaction of FA. F55F had an even higher ratio of 0.92, surpassing that of plain.

Figure 11 presents the compressive strength results of specimens cured at 43 °C over time. As expected, the 7-day compressive strength for all specimens was significantly higher compared to those cured at other temperatures. The 7-day compressive strengths at 43 °C relative to 23 °C for plain, F35, F55, and F55F were 118.3%, 138.9%, 142.4%, and 150.3%, respectively. The increase in 7-day compressive strength due to high-temperature curing was more pronounced in FA-containing specimens than in the plain specimen. Specifically, the 7-day compressive strengths of F35 and F55F were 6.2% and 8.6% higher than that of plain, respectively. At 43 °C, all FA-containing specimens demonstrated higher compressive strength than plain after 28 days. While the compressive strengths of plain and F35 increased gradually after 28 days, F55 and F55F continued to increase significantly. Notably, the 91-day compressive strength of F55F was 33.4% higher than that of plain. The effect of curing temperature on hydration can be explained by the Arrhenius equation, which indicates that temperature significantly influences the rate of chemical reactions [40,41]. As temperature increases, molecular movement intensifies, raising the average kinetic energy of molecules. This increases the likelihood of molecular collisions and bond formations, thereby accelerating the reaction rate. The higher compressive strength of FA-containing concrete at elevated temperatures is likely due to the greater temperature sensitivity of FA’s activation energy.

Figure 12 displays the compressive strength increments by curing temperature after 28 days. For the plain specimen, the highest strength increment was observed at a curing temperature of 23 °C, whereas for FA-containing specimens, the largest increment occurred at 13 °C. This trend is particularly evident in specimens with a high FA content, where the initial hydration reaction of the cement is slower, resulting in lower early-age compressive strength. However, over the long term, the pozzolanic reaction causes a significant increase in compressive strength. Generally, the activation energy, which indicates the sensitivity of cementitious materials to temperature, decreases with the addition of FA and increases with the addition of GGBS [34]. This explains why FA-blended cementitious materials are more sensitive to temperature changes than those containing GGBS [42,43].

## 4. Conclusions

This study experimentally investigated the impact of curing temperature on the hydration and mechanical properties of HVFA concrete. The key findings are summarized as follows.

Both OPC and FA cement showed a longer induction period as the curing temperature decreased, consistent with the setting time results. This effect was stronger with a higher FA content, as FA adsorbs calcium ions, lowering the Ca/Si ratio and producing calcium silicate hydrates with a reduced Ca/Si ratio.The heat of hydration decreased as the FA content increased. Hydration reactions were inhibited at low temperatures and accelerated at high temperatures, with finely ground FA providing nucleation sites, enhancing the filler effect, and shortening the induction period.Compared to the plain specimen, the higher FA content resulted in a greater decrease in compressive strength at a low temperature. The plain specimen showed a gradual strength increase after 28 days, while FA-containing specimens showed a larger increase due to the pozzolanic reaction of FA.Specimens cured at 43 °C had a significant increase in 7-day compressive strength, particularly in FA-containing specimens. Plain specimens showed slow strength increases after 28 days, but FA-containing specimens experienced substantial gains, attributed to FA’s pozzolanic reaction and temperature sensitivity.High FA content specimens had slower initial hydration and lower early strength, but the pozzolanic reaction led to significant long-term strength gains. FA’s high temperature sensitivity requires careful consideration when using HVFA concrete.FA is more temperature-sensitive than other materials, making HVFA concrete highly susceptible to changes in properties due to climate or curing temperature. Environmental factors, such as temperature, should be considered in the design and maintenance of HVFA concrete structures.

## Figures and Tables

**Figure 1 materials-17-04716-f001:**
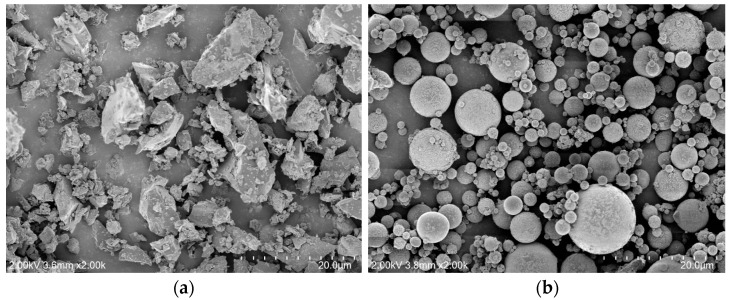
SEM images: (**a**) OPC; (**b**) FA.

**Figure 2 materials-17-04716-f002:**
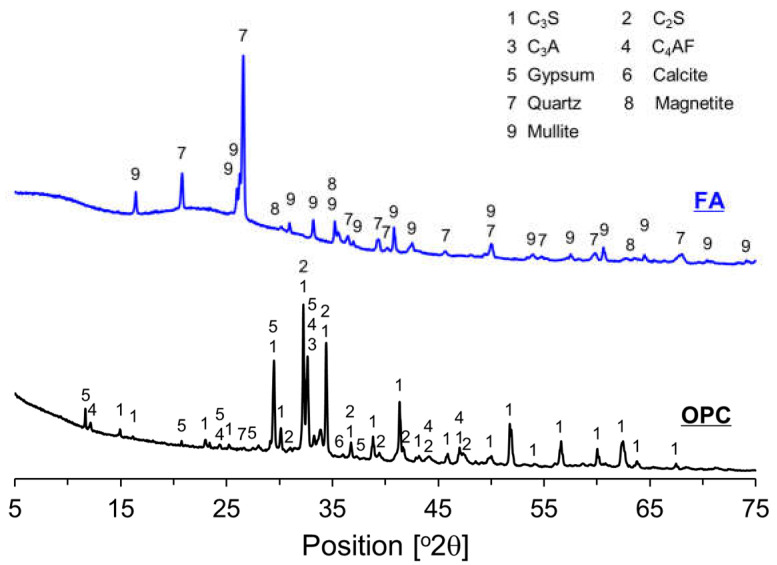
XRD patterns of OPC and FA.

**Figure 3 materials-17-04716-f003:**
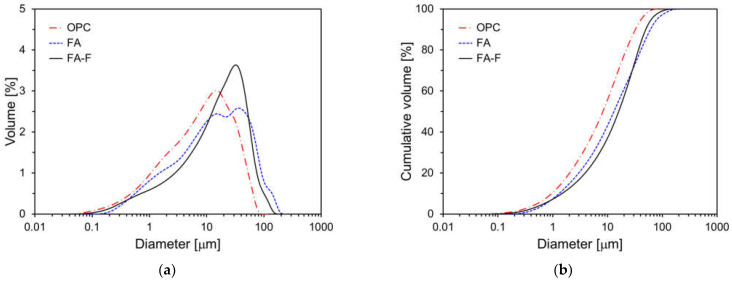
Particle size distributions of raw materials: (**a**) volume; (**b**) cumulative volume.

**Figure 4 materials-17-04716-f004:**
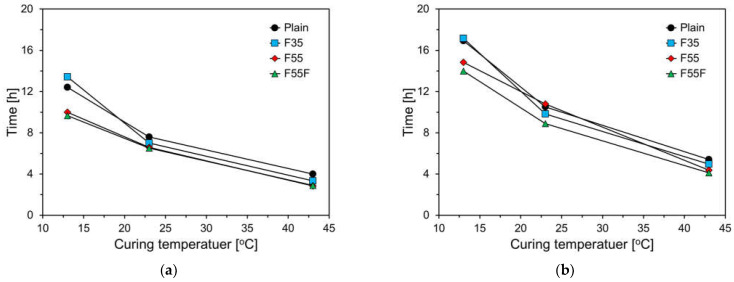
Setting times of specimens at different curing temperatures: (**a**) initial setting time; (**b**) final setting time.

**Figure 5 materials-17-04716-f005:**
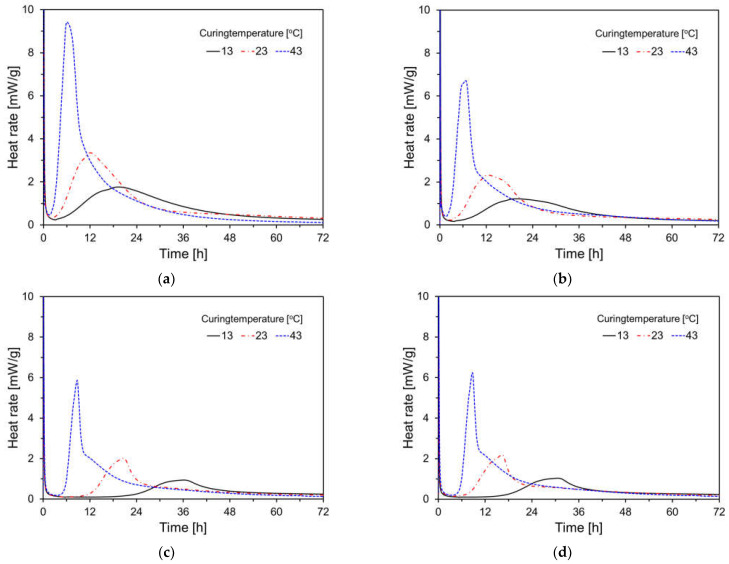
Heat rate results for specimens by curing temperature: (**a**) plain; (**b**) FC35; (**c**) FC55; (**d**) FC55F.

**Figure 6 materials-17-04716-f006:**
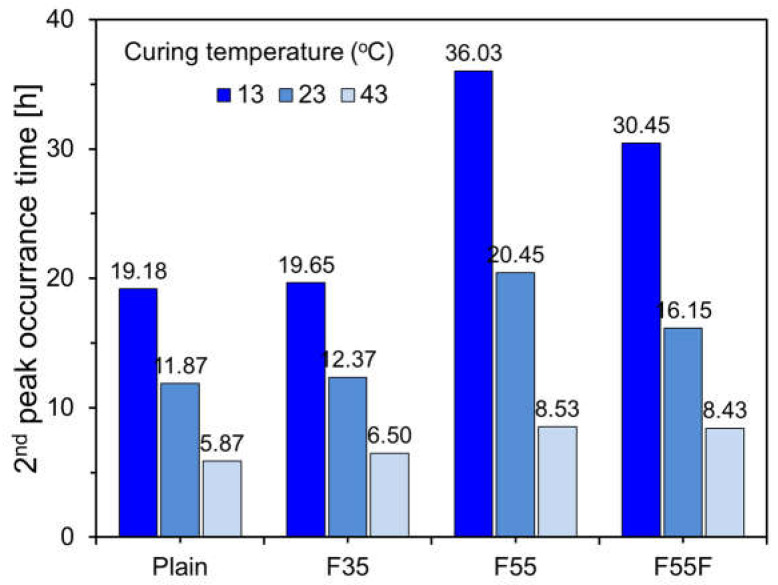
Second peak results of specimens by curing temperature.

**Figure 7 materials-17-04716-f007:**
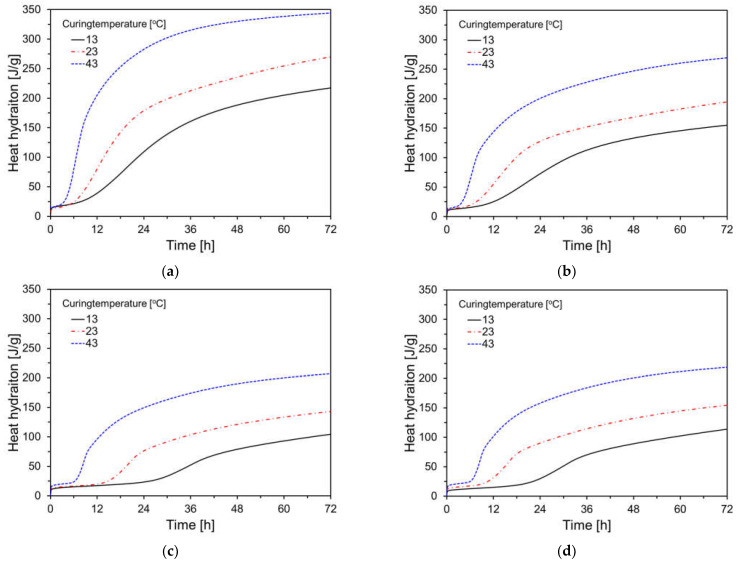
Heat hydration results for specimens by curing temperature: (**a**) plain; (**b**) FC35; (**c**) FC55; (**d**) FC55F.

**Figure 8 materials-17-04716-f008:**
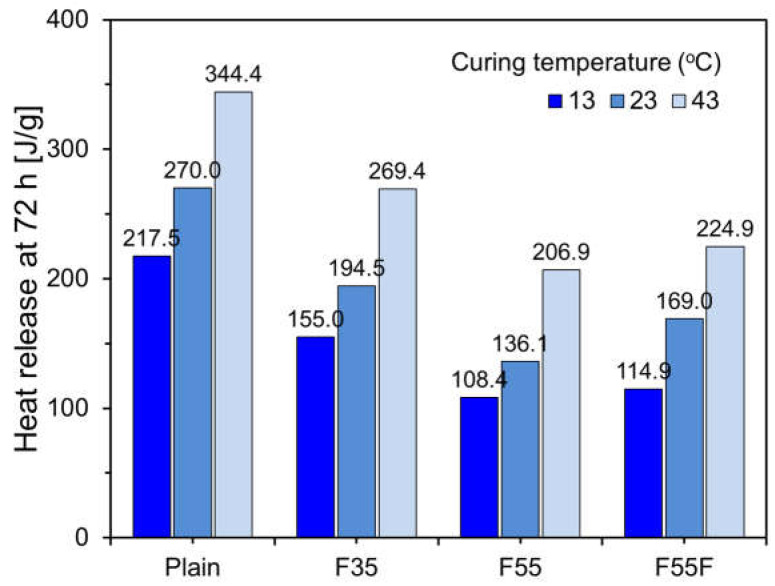
The 72 h cumulative heat release of specimens under different curing temperatures.

**Figure 9 materials-17-04716-f009:**
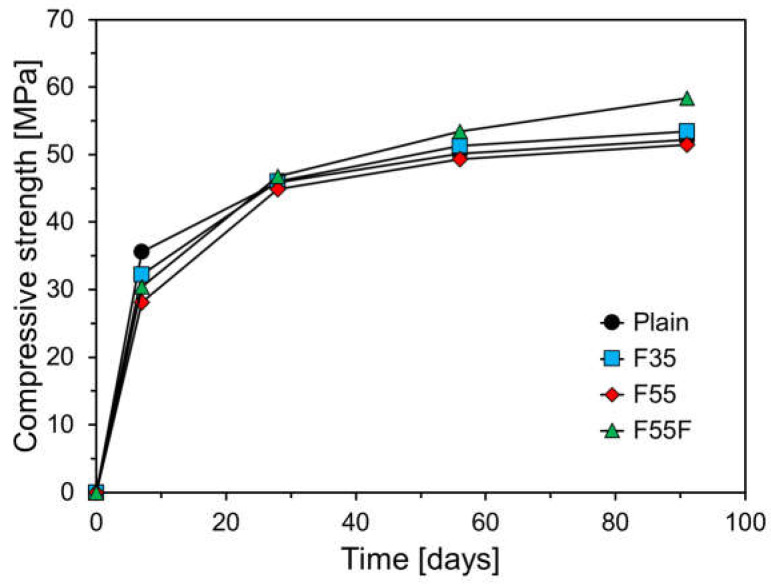
Compressive strength results at a curing temperature of 23 °C.

**Figure 10 materials-17-04716-f010:**
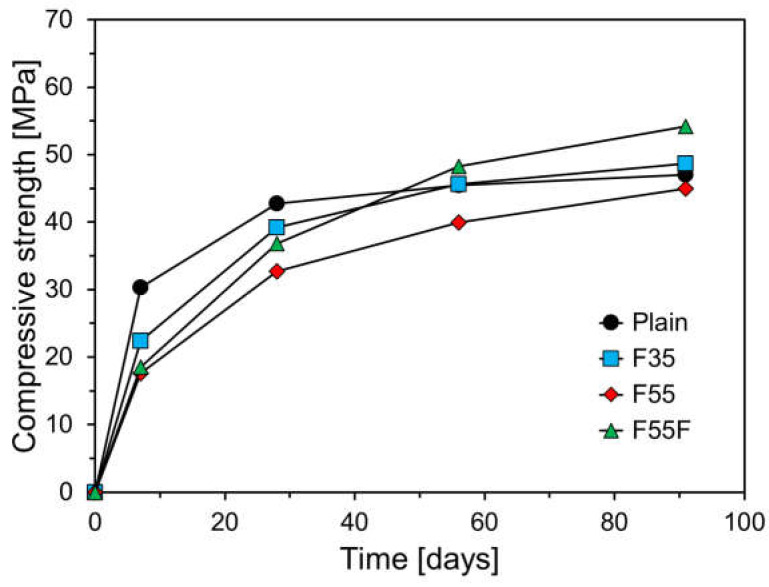
Compressive strength results at a curing temperature of 13 °C.

**Figure 11 materials-17-04716-f011:**
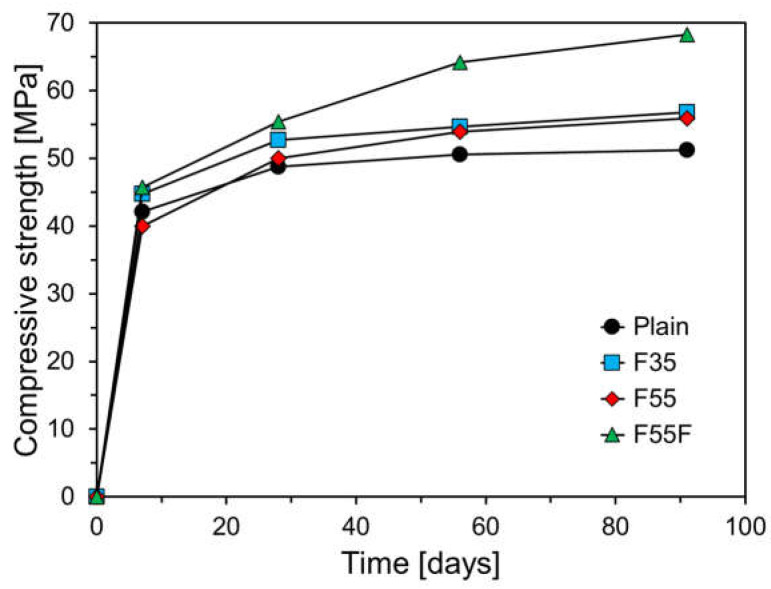
Compressive strength results at a curing temperature of 43 °C.

**Figure 12 materials-17-04716-f012:**
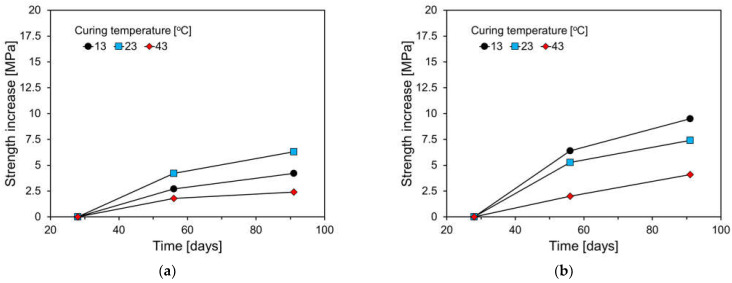
Compressive strength increment after 28 days: (**a**) plain; (**b**) FC35; (c) FC55; (**d**) FC55F.

**Table 1 materials-17-04716-t001:** Chemical compositions of OPC and FA.

	Chemical Compositions (% by Mass)								
	SiO_2_	Al_2_O_3_	Fe_2_O_3_	CaO	MgO	K_2_O	Na_2_O	SO_3_	LOI
OPC	20.8	4.6	3.7	61.8	2.9	0.8	0.4	2.3	1.1
FA	51.3	23.7	7.8	4.9	1.7	1.2	1.5	0.4	3.6

**Table 2 materials-17-04716-t002:** Mixture proportion of specimens in the experiments.

Variables	Water (kg/m^3^)	Water Binder Ratio (-)	Binder (kg/m^3^)			Aggregate (kg/m^3^)	
			OPC	FA	FA-F	Fine	Coarse
Plain	170	0.45	380	-	-	720	1033
FC35	165	0.39	273	147	-	690	989
FC55	125	0.28	200	244	-	710	1019
FC55F	125	0.28	200	122	122	710	1019

## Data Availability

The original contributions presented in the study are included in the article, further inquiries can be directed to the corresponding author.

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
