# Peer review of "Hydration and Mechanical Properties of High-Volume Fly Ash Cement under Different Curing Temperatures"

_materials, 2024, doi:10.3390/ma17194716_

Round 1

Reviewer 1 Report

Comments and Suggestions for Authors

1.    Abstract: The Abstract needs to be rewritten and better organised, following the following guidelines:

a.    Define well the objective of the work;

b.    Mention briefly the methods used in the development of the research;

c.    Mention precisely the curing ages

d.    Mention precisely the ages at which the mechanical strength tests were performed, as well as the values of these strengths

e.    Highlight at the end of the Abstract the impact produced by this research.

2.    Introduction: The Introduction is too long compared to the total length of the paper. It is recommended to summarise it.

3.    Section 2. It is recommended to change "Experimental details" to "Materials and methods".

4.    Lines 104 to 106. It is not clear when the author writes: ‘’The FA 104 contains 4.9 % CaO and 82.8 % combined SiO2, Al2O3, and Fe2O3…”. It causes some confusion. It is recommended to write as follows: “The FA contains CaO: 4.9%, SiO2: 51.3%, Al2O3: 23.7% and Fe2O3: 7.8%...”

5.    Lines 103 - 104: The sentence “The elemental oxide contents of the OPC and fly ash, determined by XRF...” is incorrect. It is better to write it as “The oxide compounds were determined by XRF...”

6.    Figure 3 appears before Figure 2. It has to be corrected.

7.    It is recommended that the author open a new sub-section entitled: 2.2. ‘Methods’, in which he explains in detail the SEM and XRD methods used in this work. Additionally, Figures 1, 2 and 3 should be included in the ‘Results’ section.

8.    Line 175. The author writes: “Figure 3 displays the initial and final setting times…”. This is an error, it actually refers to Figure 4.

9.    Attention!: There are two Figures with the number 3. See Lines 124 and 219.

10. Figures 6 and 10. Please translate the figures caption into English.

Reviewer 2 Report

Comments and Suggestions for Authors

In the reviewed paper, the authors presented research on high-volume fly ash (HVFA) concrete. The findings seem significant in the context of sustainable development and reduction of carbon dioxide emissions in the construction industry. Fly ash, as a byproduct, represents an effective way to reduce the negative environmental impact of Portland cement. Despite its advantages, there is a lack of research on the impact of curing temperature on the mechanical and hydraulic properties of HVFA concrete. The authors of the reviewed manuscript focused on analyzing studies concerning the effect of curing temperature on HVFA concrete properties, taking into account experimental results and their implications for engineering practice. Below are some comments to improve the reviewed work:

- The authors provided extensive context regarding the need to reduce CO2 emissions and the use of fly ash. The reviewer suggests that it would be beneficial to describe in more detail why studying the impact of temperature on HVFA concrete is important. What are the gaps in current research, and what specific problems are the authors trying to address?

- The description of the materials used is quite detailed. In the reviewer’s opinion, it would be useful to add information about the origin and specifications of the fly ash and cement in the context of local or international standards. What are the typical values for these materials, and how do the materials used in this study align with these standards?

- In the research methods, particularly in the section on calorimetry and strength tests, it would be valuable to clarify how experimental conditions were controlled. Were there any procedures to ensure the repeatability of the results?

- The authors present the research results in considerable detail. The reviewer notes the potential reasons behind the observed trends. For example, what chemical or physical mechanisms might explain the differences in heat of hydration and strength depending on temperature and fly ash content?

- Comparing the results with other studies would be helpful to confirm or challenge the findings. Are the observed effects consistent with the literature? What are the differences, and why might they occur?

- Were the results subjected to statistical analysis? What statistical tests were applied to ensure that the observed differences are statistically significant?

- What are the practical implications of the results for the construction industry? What are the limitations of the study, and how can they be addressed in future research?

- The conclusions should include specific recommendations for the use of HVFA concrete under different curing temperature conditions. Addressing these comments in the reviewer’s opinion will help make the manuscript more comprehensive and useful both in scientific and practical contexts.

Comments on the Quality of English Language

 Minor editing of English language required.

Reviewer 3 Report

Comments and Suggestions for Authors

The topic of this paper meets the requirements of materials, but there are some problems. The details are as follows: 1. There are many researches on fly ash at present, so how to distinguish this paper from other researches. 2. The number of samples and data processing methods are not clearly expressed. 3.134-137 line spacing is different from other parts and needs to be checked.

 4. Lines 187 needs to be checked. 5. Some languages are not English. 6. The conclusion is more concise.

Round 2

Reviewer 1 Report

Comments and Suggestions for Authors

The authors have answered all questions asked and have been responsive to the recommendations suggested.

Reviewer 3 Report

Comments and Suggestions for Authors

The revised manuscript has been improved and meets the requirements of special Issue (Low-Carbon Building Materials), so it is recommended to accept the revised manuscript.